# Solid Phase Synthesis and TAR RNA-Binding Activity of Nucleopeptides Containing Nucleobases Linked to the Side Chains via 1,4-Linked-1,2,3-triazole

**DOI:** 10.3390/biomedicines12030570

**Published:** 2024-03-03

**Authors:** Piotr Mucha, Małgorzata Pieszko, Irena Bylińska, Wiesław Wiczk, Jarosław Ruczyński, Katarzyna Prochera, Piotr Rekowski

**Affiliations:** 1Laboratory of Biologically Active Compounds Chemistry, Department of Molecular Biochemistry, Faculty of Chemistry, University of Gdansk, Wita Stwosza 63, 80-308 Gdansk, Poland; gosia.pieszko@gmail.com (M.P.); jaroslaw.ruczynski@ug.edu.pl (J.R.); katarzyna.prochera@phdstud.ug.edu.pl (K.P.); piotr.rekowski@ug.edu.pl (P.R.); 2Environmental Nucleic Acid Laboratory, Faculty of Chemistry, University of Gdansk, Wita Stwosza 63, 80-308 Gdansk, Poland; 3Laboratory of Photobiophysics, Department of Biomedical Chemistry, Faculty of Chemistry, University of Gdansk, Wita Stwosza 63, 80-308 Gdansk, Poland; irena.bylinska@ug.edu.pl (I.B.); wieslaw.wiczk@ug.edu.pl (W.W.)

**Keywords:** nucleobase amino acid, nucleopeptide, click chemistry, triazole, TAR RNA, RNA–nucleopeptide interaction

## Abstract

Nucleopeptides (NPs) represent synthetic polymers created by attaching nucleobases to the side chains of amino acid residues within peptides. These compounds amalgamate the characteristics of peptides and nucleic acids, showcasing a unique ability to recognize RNA structures. In this study, we present the design and synthesis of Fmoc-protected nucleobase amino acids (1,4-TzlNBAs) and a new class of NPs, where canonical nucleobases are affixed to the side chain of *L*-homoalanine (Hal) through a 1,4-linked-1,2,3-triazole (HalTzl). Fmoc-protected 1,4-TzlNBAs suitable for HalTzl synthesis were obtained via Cu(I)-catalyzed azide–alkyne cycloaddition (CuAAC) conjugation of Fmoc-*L*-azidohomoalanine (Fmoc-Aha) and *N1-* or *N9*-propargylated nucleobases or their derivatives. Following this, two trinucleopeptides, HalTzl_AAA_ and HalTzl_AGA_, and the hexanucleopeptide HalTzl_TCCCAG_, designed to complement bulge and outer loop structures of TAR (*trans*-activation response element) RNA HIV-1, were synthesized using the classical solid-phase peptide synthesis (SPPS) protocol. The binding between HalTzls and fluorescently labeled 5′-(FAM(6))-TAR UCU and UUU mutant was characterized using circular dichroism (CD) and fluorescence spectroscopy. CD results confirmed the binding of HalTzls to TAR RNA, which was evident by a decrease in ellipticity band intensity around 265 nm during complexation. CD thermal denaturation studies indicated a relatively modest effect of complexation on the stability of TAR RNA structure. The binding of HalTzls at an equimolar ratio only marginally increased the melting temperature (T_m_) of the TAR RNA structure, with an increment of less than 2 °C in most cases. Fluorescence spectroscopy revealed that HalTzl_AAA_ and HalTzl_AGA_, complementary to UUU or UCU bulges, respectively, exhibited disparate affinities for the TAR RNA structure (with K_d_ ≈ 30 and 256 µM, respectively). Hexamer HalTzl_TCCCAG_, binding to the outer loop of TAR_UCU_, demonstrated a moderate affinity with K_d_ ≈ 38 µM. This study demonstrates that newly designed HalTzls effectively bind the TAR RNA structure, presenting a potential new class of RNA binders and may be a promising scaffold for the development of a new class of antiviral drugs.

## 1. Introduction

The conventional depiction of RNA portrays it as an intermediary facilitating the transfer of genetic information stored in DNA. However, its role extends far beyond mere mediation. RNA is not a passive intermediary but a primary regulator associated with the flow of genetic information in the cell [1,2]. Consequently, RNA has emerged as a potential therapeutic target or tool [3,4,5]. Due to limitations in the ribose phosphate backbone of RNA, such as low cellular stability, constrained transport across cell membranes, and moderate affinity for target nucleic acid sequences, numerous efforts have been made to modify RNA’s structure to obtain ligands with improved physicochemical, biochemical, or pharmacological parameters [6,7,8]. One avenue of this research involves replacing the natural RNA backbone with classic peptide or peptide-like backbones [9,10,11,12,13]. Such biomimetics often have better physicochemical and biochemical parameters than natural nucleic acids.

While many small- and large-molecule RNA inhibitors exist, and despite their advantages, such as high specificity and affinity for recognizing RNA structures, researchers continue to seek new mimetics with even better properties [14,15,16,17,18]. Nucleopeptides (NPs) are commonly used inhibitors of RNA activity [9,10,11,19,20,21]. NPs are artificial polymers created by attaching nucleobases to the side chains of amino acid residues in peptides. These hybrid chemical structures possess the unique ability to recognize RNA. Among the well-known examples are peptide nucleic acids (PNAs) [22,23,24], which contain nucleobases connected through a short acetyl linker to a nonclassical peptide chain consisting of *N*-(2-aminoethyl)glycine residues. PNA, due to its high specificity and nucleic acid recognition affinity, is currently the most efficient type of nucleopeptide designed to date. However, it is burdened by poor solubility and a tendency to self-aggregate, especially for longer sequences [25,26]. Other limitations to the use of PNAs as nucleic acid activity-blocking ligands include their inability to enter the cell by penetrating the cell membrane [25]. However, despite their advantages, researchers are still in pursuit of new nucleic acid mimetics with improved properties. NP analogs containing the classic peptide chain are also recognized. The plasticity and ease of modification of the peptide chain mean that NPs based on the classical peptide backbone offer hope of eliminating the disadvantages of PNA [27,28,29,30]. Recently, NPs have emerged as an alternative to PNA, demonstrating the ability to penetrate the cell nucleus and exhibit antisense properties without cytotoxic effects [19,20,21,22,23,24,31,32]. The phenomenon of chirality of NPs, unlike non-chiral PNAs, can also be used to modulate their activity in nucleic acid recognition. Data available in the literature show that *L*-NPs form stable complexes with RNA, unlike their *D*-enantiomers [31]. The potential for using NPs as antiviral drugs goes much further than the inhibition of nucleic acid activity associated with viral replication [19]. Results from some experiments show that NPs block the activity of viral enzymes such as reverse transcriptase (RT), which catalyzes one of the key steps in HIV replication [29]. This suggests that these compounds may exhibit a much broader spectrum of biological activity than was assumed when designing their structures.

RNA plays a pivotal role in controlling various pathological processes, including those involved in viral replication. As a result, several RNA molecules or their fragments have garnered interest for their potential use as therapeutic targets or as model systems to understand the mechanisms of RNA recognition [33,34,35,36]. One extensively studied RNA-based model is the *trans*-activation responsive region (TAR) RNA located at the 5′-end of nascent transcripts of HIV-1, responsible for elongating proviral DNA during integration into the host chromosome. TAR RNA features a 59 bp stem–bulge–loop structure [37,38,39]. The three-nucleotide bulge (U23CU25 or U23UU25) interacts with an arginine-rich region (ARR) of the 86-amino acid Tat HIV-1 protein. Short peptides containing an isolated ARR sequence can also recognize the TAR RNA bulge [18,39,40,41]. Additionally, the six-nucleotide apical loop of the upper hairpin binds cellular proteins like transactivating factors crucial for the hyperphosphorylation of RNA polymerase II (RNAPII) and the elongation of viral transcripts [40,41,42,43,44].

TAR is one of the key regulatory structures of HIV [45]. When it is deleted or mutations are introduced into the TAR structure, especially in the looping and outer loop region, the process of transcription of proviral DNA by RNAP II is inhibited, preventing the formation of new viral particles [45]. For this reason, the TAR structure is a promising yet pharmacologically untapped target for designing inhibitors that block HIV replication at its TAR-dependent transactivation stage [45].

Designing ligands to inhibit interactions by binding to the bulge and apical loop structures of TAR RNA presents a potential opportunity to enhance the arsenal of antiretroviral compounds capable of impeding HIV replication. Numerous TAR RNA inhibitors, including NPs, have been documented in the literature; however, none are currently employed in clinical therapy [46,47,48,49,50]. Unfortunately, the high genetic variability of the virus necessitates an ongoing search for new chemical compounds adept at recognizing mutating sequences within the viral genetic material. The limited understanding of the molecular rules governing the RNA recognition process prompted the design of a new class of ligands that could potentially serve as inhibitors of HIV replication. Capitalizing on the popularity and success of the concept of click chemistry, along with the biological compatibility of the triazole moiety and the nucleobases’ ability to recognize complementary sequences, we designed and synthesized Fmoc-protected 1,4-TzlNBAs and HalTzls. These structures contain canonical nucleobases attached to the side chains via a 1,4-linked-1,2,3-triazole linker. The data on the design and synthesis of 1,4-TzlNBAs and HalTzls, along with their ability to bind to the structure of TAR RNA, are presented.

## 2. Materials and Methods

All reagents were utilized as purchased from commercial suppliers without additional purification. The progression of the reactions was monitored using TLC with Merck precoated aluminum plates 60 F_254_ (Merck Life Science, Poznan, Poland) featuring a 0.2 mm layer of silica gel containing a fluorescence indicator. Analytical RP HPLC analyses were conducted on a Beckman instrument (Beckman Instruments, Fullerton, CA, USA) using a Phenomenex Kinetex C18 column (4.6 mm × 150 mm, 5 µm particle size). Preparative HPLC purifications were performed on a Spot Prep II 250 instrument (Armen Instrument, Saint-Ave, France) utilizing a C18 reversed-phase column. The ^1^H and ^13^C measurements were carried out in DMSO-d_6_ using a Bruker AVANCE III 500 MHz spectrometer (Bruker Daltonics, Billerica, MA, USA) and standard Bruker software (Analyst TF 1.7.1.); δ in parts per million; J in hertz. Mass spectra were recorded on a triple quadrupole time-of-flight (QTOF) mass spectrometer AB SCIEX 5600+ (AB Sciex LLC, Framingham, MA, USA) equipped with a duo-electrospray interface and operated in a positive ionization mode.

### 2.1. General Procedure for the Synthesis of the Monopropargyl Nucleobases (5)

*N1*-propargylthymine, *N4*-acetyl-*N1*-propargylcytosine, *N9*-propargyladenine, and *N9*-propargylguanine were obtained by reacting thymine, *N4*-acetylcytosine, adenine, and 2-amino-6-chloropurine with propargyl bromide in the presence of potassium carbonate in DMF at room temperature, as described in the literature [51]. The mixture of 2 mmol of the heterocyclic base (thymine, *N4*-acetylcytosine, adenine, 2-amino-6-chloropurine), 1.9 mmol of K_2_CO_3_, and 2 mmol of propargyl bromide in 20 mL of anhydrous DMF was stirred at room temperature for 24 h. After removing the solvent under reduced pressure, the residue obtained was purified on a silica gel column (DCM:MeOH, 95:5 *v*/*v*).

*N1*-propargylthymine (6) was obtained as a pale yellow solid (91.93 mg, 56%). ESI-MS: *m*/*z* calcd and found for [M+H]^+^ 165.06.

*N4*-acetylcytosine (10) was obtained as a white solid (220 mg, 72%). ESI-MS: *m*/*z* calcd for [M+H]^+^ 154.01; found: 154.03.

*N4*-acetyl-*N1*-propargylcytosine (7) was obtained as a white solid (120 mg, 85%). ESI-MS: *m*/*z* calcd for [M+H]^+^ 192.08; found: 192.07.

*N9*-propargyladenine (8) was obtained as a white solid (98.71 mg, 57%). ESI-MS: *m*/*z* calcd and found for [M+H]^+^ 174.07.

*2*-amino-6-chloro-N9-propargylpurine (12) was obtained as a pale yellow solid (311 mg, 75%). ESI-MS: m/z calcd and found for [M+H]^+^ 208.03.

*N9*-propargylguanine (9): A suspension of 12 (190.00 mg, 0.915 mmol) was dissolved in 1 M HCl (4 mL). The solution was heated at reflux for 1 h. After cooling, the solution was neutralized to pH 7 with 10% NaOH. The solvent was removed under reduced pressure, and the residue was purified by preparative C18 RP-HPLC to yield **5** as a white solid (167.9 mg, 97%). ESI-MS: *m*/*z* calcd and found for [M+H]^+^ 190.07.

### 2.2. General Procedure for the Synthesis of 1,4-TzlNBAs

To a mixture of the corresponding *N*-propargyl nucleobase (1 equiv) and Fmoc-Aha (1.2 equiv) in H_2_O/*t*-BuOH (1:1), a 100 mM solution of CuSO_4_ × 5H_2_O (1.2 equiv) and a freshly prepared 500 mM solution of sodium ascorbate (4 equiv) were added. The reaction mixture was stirred at room temperature for 1 day and monitored by RP-HPLC. After completion, the resulting slurry was filtered, and the solvent was removed under reduced pressure. The crude product was pre-purified by precipitation from DMSO with cold water. A preparative C18 RP HPLC was then applied.

2-(Fmoc-amino)-4-[4-(*N1*-methylthymine)-*1H*-1,2,3-triazol-1-yl)butanoic acid (1)

According to the general procedure, from alkyne 6 (100 mg, 0.609 mmol) and Fmoc-Aha (267.8 mg, 0.731 mmol), monomer 1 (239.09 mg, 74%) was obtained after a preparative C18 RP-HPLC purification as a white solid. ESI-MS: m/z calcd for [M+H]^+^ 531.19; found: 531.15.

^1^H NMR (DMSO-d6, δ): 12.73 (1H, COOH, s); 11.23 (1H, NH(T), s); 7.98 (1H, H5(Tzl), s); 7.83 (2H, Fmoc, d, J = 7.55 Hz); 7.73 (1H, NH(Hal), d, J = 8.25 Hz); 7.66 (2H, Fmoc, d, J = 7.4 Hz); 7.53 (1H, H6, s); 7.35 (2H, Fmoc, ddd, J = 2.9 Hz); 7.27 (2H, Fmoc, m); 4.82 (2H, Tzl-CH_2_-T, s); 4.32 (2H, Tzl-***CH_2_***-CH_2_, m); 4.26 (2H, OCH_2_, d, J = 7.05 Hz); 4.18 (1H, CH(Fmoc), t, J = 6.8 Hz); 3.91 (1H, CH(Hal), m); 2.23 i 2.03 (2H, Tzl-CH_2_-***CH_2_***, m); 1.68 (3H, CH_3_, s).

^13^C NMR (DMSO-d6, δ): 173.50; 164.73; 156.61; 151.19; 144.27; 143.01; 141.58; 141.21; 128.12; 127.57; 125.69; 124.07; 120.59; 109.32; 66.10; 51.66; 47.13; 46.97; 42.65; 31.77; 12.41.

2-(Fmoc-amino)-4-[4-(*N1*-methyl-*N4*-acetylcytosine)-*1H*-1,2,3-triazol-1-yl)butanoic acid (2)

According to the general procedure, from alkyne ***7*** (100 mg, 0.523 mmol) and Fmoc-Aha (230 mg, 0.628 mmol), monomer 2 (201.2 mg, 69%) was obtained after precipitation from DMSO with cold water and purification of the resulting precipitate using preparative C18 RP HPLC as a white solid as a white solid. ESI-MS: m/z calcd for [M+H]^+^ 558.21; found: 558.25.

^1^H NMR (DMSO-d6, δ): 12.81 (1H, COOH, s); 10.84 (1H, NH(C), s); 8.19 (1H, H6, d, J = 7.25 Hz); 8.07 (1H, H5(Tzl), s); 7.90 (2H, Fmoc, d, J = 7.55 Hz); 7.80 (1H, NH(Hal), d, J = 8.05 Hz); 7.73 (2H, Fmoc, d, J = 7.45 Hz); 7.42 (2H, Fmoc, ddd, J = 3.35 Hz); 7.34 (2H, Fmoc, m); 7.17 (1H, H5, d, J = 7.25 Hz); 5.06 (2H, Tzl-CH_2_-C, s); 4.39 (2H, Tzl-*CH_2_*-CH_2_, m); 4.33 (2H, OCH_2_, m); 4.25 (1H, CH(Fmoc), t, J = 6.9 Hz); 3.92 (1H, CH(Hal), m); 2.30 (2H, Tzl-CH_2_-*CH_2_*, m); 2.09 (3H, CH_3_(Ac), s).

^13^C NMR (DMSO-d6, δ): 173.50; 171.37; 162.98; 156.61; 155.42; 150.60; 144.26; 141.21; 128.12; 127.57; 125.69; 124.46; 120.61; 95.78; 66.10; 51.68; 47.05; 44.87; 31.72; 24.78.

2-(Fmoc-amino)-4-[4-(*N9*-methyladenine)-*1H*-1,2,3-triazol-1-yl)butanoic acid (3)

According to the general procedure, from alkyne **8** (173 mg, 1 mmol) and Fmoc-Aha (439.7 mg, 1.2 mmol), monomer 3 (415.45 mg, 77%) was obtained after precipitation from DMSO with cold water and purification of the resulting precipitate using preparative C18 RP HPLC as a white solid. ESI-MS: m/z calcd for [M+H]^+^ 540.21; found: 540.16.

^1^H NMR (DMSO-d6, δ): 12.81 (1H, COOH, s); 8.26 i 8.21 (2H, H2 i H8, 2s); 8.09 (1H, H5(Tzl), s); 7.90 (2H, Fmoc, d, J = 7.4 Hz); 7.79 (1H, NH(Hal), d, J = 8.15 Hz); 7.73 (2H, Fmoc, d, J = 5.5 Hz); 7.61 (2H, NH_2_, s); 7.42 (2H, Fmoc, ddd, J = 4.15 Hz); 7.33 (2H, Fmoc, dd, J = 6.4 Hz); 5.46 (2H, Tzl-CH_2_-A, s); 4.38 (2H, Tzl-***CH_2_***-CH_2_, m); 4.33 (2H, OCH_2_, d, J = Hz); 4.25 (1H, CH(Fmoc), t, J = 6.9 Hz); 3.91 (1H, CH(Hal), m); 2.30 i 2.11 (2H, Tzl-CH_2_-***CH_2_***, m).

^13^C NMR (DMSO-d6, δ): 173.49; 156.60; 151.60; 149.60; 144.26; 142.84; 141.21; 128.12; 127.56; 125.70; 124.17; 120.61; 66.09; 51.63; 47.06; 38.64; 31.67.

2-(Fmoc-amino)-4-[4-(*N9*-methylguanine)-*1H*-1,2,3-triazol-1-yl)butanoic acid (4)

According to the general procedure, from alkyne 9 (100 mg, 0.529 mmol) and Fmoc-Aha (232.6 mg, 0.635 mmol), monomer 4 (211.59 mg, 72%) was obtained after precipitation from DMSO with cold water and purification of the resulting precipitate using preparative C18 RP HPLC as a white solid. ESI-MS: m/z calcd for [M+H]^+^ 556.20; found: 556.09.

^1^H NMR (DMSO-d6, δ): 12.83 (1H, COOH, s); 10.91 (1H, NH(G), s); 8.19 (1H, H8, s); 8.06 (1H, H5(Tzl), s); 7.90 (2H, Fmoc, d, J = 7.55 Hz); 7.80 (1H, NH(Hal), d, J = 8.15 Hz); 7.73 (2H, Fmoc, d, J = 6.05 Hz); 7.42 (2H, Fmoc, ddd, J = 3.85 Hz); 7.34 (2H, Fmoc, q, J = 6.95 Hz); 6.69 (2H, NH_2_(G), s); 5.30 (2H, Tzl-CH_2_-G, s); 4.40 (2H, Tzl-***CH_2_***-CH_2_, m); 4.34 (2H, OCH_2_, d, J = 7 Hz); 4.25 (1H, CH(Fmoc), t, J = 6.9 Hz); 3.91 (1H, CH(Hal), m); 2.30 i 2.11 (2H, Tzl-CH_2_-***CH_2_***, m).

^13^C NMR (DMSO-d6, δ): 173.50; 156.63; 154.80; 144.25; 141.21; 137.40; 128.12; 127.56; 125.69; 124.19; 120.60; 92.90; 66.11; 51.62; 47.12; 42.70; 38.60; 31.50.

### 2.3. Solid-Phase Synthesis of HalTzls

Nucleopeptides HalTzls were synthesized as C-terminal amides using a semi-automatic peptide synthesizer, Peptide Synthesizer SP650 Labortec AG (Bubendorf, Switzerland). The classical SPPS method with the Fmoc strategy was used [50]. Fmoc-protected 1,4-TzlNBA monomers were assembled on TentaGel S RAM resin (capacity 0.22 mmol Fmoc-NH/1g resin; 50 mg resin was used) as active derivatives in a 3-fold molar excess of coupling reagents (Fmoc-1,4-TzlNBA: HATU: HOAt: colidine: DMAP, 1:1:1:2:0.001 molar ratio) in the *N*,*N*-dimethylformamide (DMF) solution for 60 min. Each coupling process was repeated using a 1.5-fold excess of reagents for 60 min. The removal of the Fmoc groups was carried out with a 20% piperidine/DMF in two cycles lasting 5 and 15 min, respectively. The cleavage from the resin and deprotection of the HalTzls were performed with a 98% TFA/DCM solution for 120 min. After evaporation of TFA and DCM, the crude nucleopeptide was dissolved in water and, after freezing, lyophilized. The crude HalTzls were purified using a semi-preparative RP HPLC Knauer chromatograph (Knauer, WG GmbH, Berlin, Germany) system using a Kromasil C8 column (20 × 250 mm, 10 µm particle size). A gradient of 0–60% ACN with 0.08% TFA, at a flow rate of 3.5 mL/min, was used for purification. The column was maintained at ambient temperature. The eluates were monitored with a UV detector at *λ* = 254 nm. Fractions with the highest purity (>95%) were analyzed by analytical RP-HPLC using a Beckman System Gold chromatograph (Beckman Coulter, Brea, USA) using a Kinetex C18 column (Phenomenex, 150 × 4.6 mm, 5 µm particle size) with a 0–100% ACN gradient with the addition of 0.08% TFA (solvent B) and 0.08% TFA (solvent A) in 30 min. The column was maintained at ambient temperature. The flow rate was 1 mL/min, and the eluates were monitored using a UV detector at *λ* = 254 nm. The identity of HalTzls was confirmed by mass spectrometry using a QTOF SCIEX 5600+ ESI spectrometer operating in positive mode. A summary of the physical and chemical parameters of HalTzls can be found in Table 1.

### 2.4. TAR RNA Synthesis

5′-FAM(6)-TAR RNA UCU/UUU and its unlabeled forms were obtained from FutureSynthesis (Poznan, Poland). The lyophilized samples were dissolved in TB or Tris buffer and denatured at 95 °C for 5 min before CD and fluorescence anisotropy experiments.

### 2.5. Circular Dichroism (CD) Spectropolarimetry

CD spectra were recorded using a Jasco J-815 spectropolarimeter equipped with a Peltier-controlled cell holder in a quartz cuvette with a 0.1 cm path length spanning from 200 to 310 nm at 25 °C. CD temperature-dependent experiments were conducted over the range of 25–95 °C. TAR RNA was maintained at a concentration of 5 μM for all CD experiments. To prevent changes in ellipticity values linked to potential sample dilution, the solution containing TAR RNA was introduced to the dry lyophilized HalTzl sample, resulting in a final TAR RNA concentration of 5 μM and a 10-fold excess (50 μM) of nucleopeptide concentration. Before initiating CD experiments, each sample was equilibrated for at least 30 min at the initial temperature to ensure the required temperature and achieve equilibrium in TAR—HalTzl interactions. The spectra were averaged over three scans, each recorded at a speed of 100 nm/min with a 1 nm data pitch. As HalTzls nucleopeptides exhibit a negative ellipticity effect in the 200–240 nm range, ellipticity values for TAR—HalTzls complexes were corrected for the contribution associated with HalTzls. The observed ellipticity for both free and bound (corrected ellipticity) TAR was expressed in mdeg. The baseline was corrected for every CD spectrum. Origin 6.5 software (OriginLab, Northampton, MA, USA) was utilized for the analysis of obtained data in CD melting experiments. The Boltzmann function implemented in this program was employed for fitting melting curves and calculating T_m_ values.

### 2.6. Fluorescence Anisotropy Spectroscopy

The binding affinities between HalTzls and TAR UUU/UCU structures were determined through fluorescence anisotropy measurements. This was conducted using a Fluoromax-4 spectrofluorophotometer (Horiba Jobin Yvon IBH Ltd., Kyoto, Japan) with a cylindrical quartz cuvette featuring an optical path of 3 mm. Fluorescence anisotropy was computed from the intensities detected at 518 nm with excitation at 480 nm. Each data point was measured three times with a 500 ms integration time and then averaged. A 5′-FAM(6)-TAR RNA solution (final concentration 100 nM) underwent titration with increasing amounts of HalTzl. The reactions occurred in TB buffer at 25 °C. HalTzl (ligand) solutions were prepared as serial dilutions in TB buffer. Before initiating the experiment, the FAM-labeled TAR RNA sample was refolded by heating to 90 °C for 5 min and then being allowed to cool freely to 25 °C. Subsequently, the titration was performed. After adding the ligand to TAR RNA, the sample was gently mixed by hand and incubated to achieve complexation equilibrium at 25 °C for 20 min. Binding data were analyzed using a single binding site curve-fitting procedure implemented in GraphPad Prism 3.0 software (GraphPad Software Inc., San Diego, CA, USA).

## 3. Results and Discussion

### 3.1. Synthesis of N-Propargyl Nucleobases and 1,4-TzlNBA Monomers

In the ongoing pursuit of designing and synthesizing novel NBAs, we present the synthesis of nucleobase 1,2,3-triazole amino acid derivatives (1,4-TzlNBAs) (**1–4**) featuring the complete set of canonical nucleobases connected to the side chain of the commercially available Fmoc-Aha (**13**) through a 1,4-disubstituted 1,2,3-triazole linkage (Figure 1).

The monomers (**1–4**) were conceived as building blocks for nucleobase peptides, aligning with the Fmoc strategy of SPPS. In the initial step, we synthesized *N*-propargylated nucleobase derivatives (**6–9**) as the dipolarophile components of the CuAAC reaction (Figure 1, Figure 2).

These compounds had been previously prepared using free or protected nucleobases and propargyl bromide [51]. The nucleobases were subjected to treatment with propargyl bromide in DMF in the presence of K_2_CO_3_ as a basic catalyst (Figure 1 and Figure 2). The reactions were conducted for 24 h at room temperature. Thymine and adenine were utilized as unprotected substrates, while cytosine and guanine, due to their reactive amino groups, were employed as *N4*-acetyl (**10**) and 2-amino-6-chloropurine (**11**) derivatives, respectively (Figure 2).

We chose to use *N4*-acetyl (**10**) instead of the commonly used *N4*-benzoylcytosine because propargylation of the latter resulted in *N1*-propargyl-*N4*-benzoylcytosine, which was hardly soluble in organic solvents. The acetylation of cytosine was conducted with acetic anhydride in pyridine. *N9*-propargyl guanine was obtained through 2-amino-6-chloropurine propargylation (**11**), followed by its hydrolysis (**12**) in 1 M HCl for 1 h (Figure 2). The short duration of hydrolysis was crucial, as prolonged hydrolysis led to an increased number and amount of side products. In all cases, propargylation resulted in *N1*- or *N9*-monopropargyl nucleobase as the main product of the reaction. This shows that the bromoalkylation reaction of nitrogenous bases with 1,2-dibromoethane is regioselective, albeit for some nucleobases, due to the reactivity of their functional groups, semipermanent or complete protection is necessary. In the final step, CuAAC was applied to “click” *N*-propargyl nucleobase and the azido moiety of Fmoc-*L*-Aha (**13**) (Figure 3).

The four target Fmoc-1,4-TzlNBAs (**1–4**) were successfully prepared. The reaction was carried out overnight in an H_2_O/*t*-BuOH solution using CuSO_4_ and a freshly prepared sodium ascorbate solution. The final products were purified by column chromatography, and their structures were characterized by ^1^H and ^13^C NMR. Since none of the reactions used in the synthesis of the NBA monomers and HalTzls oligomers resulted in a change in the configuration of the asymmetry center (α-carbon atom) of the starting compound (Fmoc-*L*-Aha), all the synthesized compounds (Figure 1, Figure 4) retained the *L*-configuration of the asymmetry center (or centers). The resulting Fmoc derivatives 1-4 are soluble in DMF, making it possible to use the standard solid-phase peptide synthesis (SPPS) protocol commonly used for the synthesis of peptides for the synthesis of HalTzls [52].

### 3.2. HalTzls Synthesis

After the synthesis of NBA monomers, we proceeded to synthesize HalTzls nucleopeptides using the SPPS method with the Fmoc strategy (Figure 4).

To mitigate the unfavorable impact of electrostatic repulsion between the negatively charged phosphate groups in the TAR RNA structure and the C-terminal carboxylate group of HalTzls nucleopeptides, the latter were synthesized in the form of C-terminal amides. Various combinations of NBA coupling conditions were tested to optimize synthesis efficiency. The best results were achieved using a dual coupling mixture of Fmoc-1,4-TzlNBA:HATU:HOAt:collidine:DMAP (1:1:1:2:0.001 molar ratio) in DMF for two 60 min cycles. After separation from the resin and purification by RP HPLC, HalTzls were obtained with moderate to good yields, ranging from about 17% to 37% (Table 1). Only in the case of HalTzl_TCCCAG_ did the cleavage procedure from the resin prove insufficient, resulting in partial removal of the protective acetyl groups located at the C4 position of the cytosine residues (Appendix A). Therefore, in this case, an additional procedure involved dissolving and leaving the resin-cleaved crude HalTzl_TCCCAG_ in a 1 M K_2_CO_3_/MeOH solution for 1 h. After this time, all acetyl protective groups of cytosine were removed (Appendix A).

Following the purification and physicochemical characterization of HalTzls (Table 1, Appendix A), their ability to bind complementary RNA sequences using 5′-FAM(6)-TAR UCU/UUU as a model system of pharmacological relevance was assessed (Figure 3). For CD experiments, an unlabeled version of TAR UCU/UUU was used.

### 3.3. CD Study of TAR RNA—HalTzls Interactions

The CD method is a widely employed technique for studying conformational changes in both free and bound RNA [53,54]. In this study, we utilized this method to characterize the behavior of TAR UCU and UUU structures during the binding of HalTzls ligands. Temperature-dependent CD studies were employed to characterize both uncomplexed and bound TARs. CD spectra of RNA (form A) typically exhibit a negative minimum around 210 and 235 nm and a positive maximum at 265 nm [54,55]. The latter serves as a diagnostic band for ligand binding, as its intensity and position are sensitive to changes in the nucleobase stacking profile. These changes correlate with conformational alterations in RNA induced by physicochemical factors (e.g., temperature) or ligand binding.

The CD spectrum of free TAR UCU and TAR UUU recorded in the TB buffer is similar for both and is characteristic of the A form of RNA (Figure 4A).

The spectra exhibit three extremes: a minimum at about 208 nm and 235 nm and a maximum at 265 nm. In the case of TAR UCU, the intensity of the extrema was slightly lower (except for 235 nm) compared to the TAR UUU variant (Figure 4A). The CD experiment conducted for TAR UUU revealed that the intensity of the ellipticity signal is sensitive to environmental parameters, such as buffer type. The CD curves of TAR UUU recorded in Tris and TB buffers differ slightly in intensity at the extreme points of these curves, likely reflecting the small conformational changes occurring in the single-stranded bulge and apical loop fragments (Figure 4C).

To assess the stability of TAR’s hairpin-bulge structure, temperature-dependent CD measurements were conducted. CD curves recorded in the range of 25–95 °C showed that the structure of both TAR variants was temperature-dependent (Figure 4B). A gradual decrease in the intensity of both extremes correlated with increasing temperature. In the case of the maximum, a shift of the positive extremum toward longer wavelengths was observed. These changes reflect conformational changes in single-stranded fragments, the bulge, and the apical loop, as well as the unraveling of double-stranded helix fragments located both above and below the bulge. Temperature-dependent changes in ellipticity at a maximum of around 265 nm were used to characterize the stability of unbound TAR structures by determining their melting temperatures (T_m_). TAR’s CD melting profiles are shown in Figure 4B,D. The curves were characterized by the distinctive sigmoid-like shape typical of unraveling double-stranded TAR fragments. The determined T_m_ values indicated that the structure of TAR UUU is much more stable in TB than in Tris buffer (T_m_ 70.41 ± 2.05 vs. 62.62 ± 0.74 °C) (Table 2). The difference is almost 4 °C. This highlights the crucial role of environmental conditions (e.g., buffer) in RNA activity studies. The greater stability of a TAR structure containing a U-rich UUU bulge relative to one containing a UCU bulge has been observed previously [56]. Our temperature-dependent CD studies have shown that the observed changes in ellipticity signal intensity, especially around the extremum at 265 nm, correlate with conformational changes affecting the stability of TAR structures.

Subsequently, CD was employed to characterize the binding of HalTzls ligands to the TAR UCU/UUU structure. As the quantitative characterization of the target interactions was determined by dissociation constants (K_d_) using fluorescence anisotropy spectroscopy, CD experiments focused on the analysis of complexes formed under conditions of a 10-fold excess of HalTzls ligands relative to TAR. In general, the CD curves illustrating the interaction of TAR with HalTzls ligands are characterized by a reduction in the intensity of the band ellipticity at 265 nm and a less intense effect for the band at 210 nm (Figure 5, Figure 6 and Figure 7). Since the HalTzls ligands themselves exhibit a negative ellipticity value in the 200–240 nm range, the CD curves of the complexes were corrected for this effect. CD curves recorded for the TAR UUU—HalTzl_AAA_ complexes showed similar binding effects in both Tris and TB buffers (Figure 5).

In both cases, the decrease in ellipticity intensity for the band at approximately 265 nm was approximately 15%. However, some differences related to the stability of the complex associated with the T_m_ parameter were observed (Figure 5, Table 2). With increasing temperature, CD melting curves were characterized by a gradual decrease in the intensity of the CD signal at a maximum of 265 nm and a shift of this maximum toward longer wavelengths. At the highest temperature investigated (95 °C), the CD maximum was moved to about 275 nm (Figure 5A,B). CD melting curves show that the stability of the complex TAR UUU—HalTzl_AAA_ characterized by the T_m_ value is higher in the TB than in the Tris buffer. These differences are derived from the primary effect of differences in the stability of unbound TAR UUU in these buffers (Figure 4B). A small difference in the T_m_ parameter between the unbound and bound form of TAR UUU was recorded in the Tris buffer (Table 2). In the TB buffer, these values are almost the same. These may indicate a relatively low affinity of HalTzl_AAA_ for the TAR UUU bulge. The CD spectrum revealed that the interaction between TAR UCU and the HalTzl_AGA_ ligand, complementary to the bulge structure, caused only a 4% decrease in ellipticity intensity at 265 nm (Figure 6).

This effect was significantly smaller, almost four times, compared to the observed effect for TAR UUU interacting with HalTzl_AAA_. Analysis of the CD melting curves indicated that this change correlated with a slight increase in stability (approximately 1 °C) of the bound TAR UCU structure (Table 1). For the TAR UCU interaction with HalTzl_TCCCAG_, complementary to the apical loop of TAR UCU, almost a 27% decrease in ellipticity intensity at 265 nm was observed (Figure 7). This effect was correlated with a 2 °C increase in the stability of the bound vs. free TAR UCU structure.

### 3.4. Fluorescence Anisotropy Study of TAR RNA—HalTzls Interactions

Fluorescence anisotropy was employed to quantify TAR—HalTzl interactions. 5′-FAM(6)-labeled TAR UUU/UCU RNA fragments were utilized as target molecules to characterize the binding properties of HalTzls ligands. The binding experiments involved measuring the decreasing fluorescence anisotropy of FAM-TAR caused by the addition of increasing amounts of HalTzls (Figure 8).

The obtained data were used to determine binding curves. The dissociation constants (K_d_) of the FAM-TAR—HalTzls complexes were then calculated (Table 3). To derive the binding curves and K_d_ values, it was assumed that the interaction corresponds to the one-binding-site model, where the HalTzl ligand binds to the complementary sequences of the bulge or the apical loop regions of TAR.

The analysis of TAR UUU interactions with HalTzl_AAA_ revealed a medium-strength binding characterized by a dissociation constant (K_d_) of approximately 30 µM (Figure 8A, Table 3). In contrast, when the HalTz_AGA_ ligand bound to the TAR UCU structure, the binding was nearly 9-fold weaker compared to TAR UUU—HalTzl_AAA_ (Figure 8B, Table 3), with a K_d_ of ~256 µM. This behavior of the HalTzl_AGA_ ligand, despite the presence of a guanine (G) residue capable of forming three hydrogen bonds (compared to the two formed by adenine (A)), can only be explained by the steric constraints present during the interaction. This is intriguing because, in both TAR UUU and UCU variants, the bulges have a high degree of conformational freedom and should easily adapt to the structure of the ligands. For TAR UCU interaction with HalTzl_TCCCAG_, a much stronger binding was observed with a K_d_ of ~38 µM compared to the interaction with the HalTzl_AGA_ ligand (K_d_ ~256 µM) (Figure 8C, Table 3). In these cases, an almost 7-fold difference in the strength of the ligand binding by the TAR structure was observed. This indicates that the HalTzl_TCCCAG_ hexamer binds much more tightly to the apical loop of TAR UCU than the HalTzl_AGA_ trimer to the UCU bulge. Although the ligand and the apical loop and bulge sequences are complementary to each other, the ligand sequences differ in length (HalTzl_TCCCAG_ sequence is 2-fold longer than that of HalTzl_AGA_). Surprisingly, the interaction strength for the TAR UCU—HalTzl_TCCCAG_ system is similar to that observed for the TAR UUU—HalTzl_AAA_. This may indicate that, in addition to the complementarity of nucleosides connected with the ability to form hydrogen bonds, other factors like interactions of nucleobases ring systems with triazole moieties of side chains of the ligands play an important role in the TAR—HalTzl interactions. The pronounced differences observed in the binding of ligands of the same size trimers HalTzl_AAA_ and HalTzl_AGA_ to the UCU bulge suggest that steric factors may also play an important role in the recognition of TAR structure by HalTzl ligands. The tested ligands demonstrated moderate (approximately 30–40 µM) or low (approximately 260 µM) affinity for TAR structures. This implies that interactions, such as hydrogen bonding or hydrophobic interactions involving complementary nucleobases and triazole rings, are insufficient to yield ligands with high affinity for TAR structures. Introducing positively charged amino acid residues, such as lysine (Lys) or arginine (Arg), into newly designed structures based on HalTzl sequences would be a reasonable strategy. This approach, commonly used in ligand design, including nucleopeptides binding to RNA/DNA, involves additional electrostatic interactions between the positively charged side chains of Lys or Arg and the negatively charged phosphate groups of the sugar–phosphate chain of RNA. Such modifications are expected to significantly increase the affinity of the HalTzl ligands for the TAR structure.

The popularity of click chemistry, especially the CuAAC variant, and the ease of formation of the triazole ring have made the design and synthesis of new systems containing this system an important branch of synthetic organic chemistry [57]. In many cases, the newly obtained triazole derivatives of known chemical structures showed greater biological activity than their unmodified counterparts [57,58,59]. In most cases, the triazole ring was introduced into small-molecule compounds, although examples of the introduction of this system into large macromolecules such as peptides or nucleic acids are known [60,61,62]. The molecular basis of such modifications underlying the changes in physicochemical properties and biological activity of such modified biomolecules is poorly understood. Previous results have shown that the presence of the triazole system strongly influences the structure and stability of nucleic acid duplexes [63]. The effect of triazole-modified 2′-deoxyuridines on the stability of DNA/DNA and DNA/RNA duplexes has been reported [64,65]. Authors observed that although a single triazole ring incorporation decreased DNA/DNA duplex stability, the stacking of a few (2–4 triazole residues) consecutive modifications led to increased stability due to additional base stacking interactions of triazole moieties. This demonstrates the complex nature of the interactions of the triazole system with the structure of the nucleic acid chain, both nucleobases and the backbone, which are likely to involve electrostatic and hydrophobic interactions, including base stacking, the possibility of forming hydrogen bonds, as well as steric hindrance, as the triazole ring is a flat aromatic system similar in size to the pyrimidine nucleobase. To our knowledge, the triazole system has not yet been introduced as a structural element in NPs. Our strategy of introducing the triazole system into each side chain and linking the nucleobase to the rest of the amino acid molecule through the triazole ring is the first step towards understanding the influence of the triazole ring on the process of nucleic acid recognition by such modified NPs.

## 4. Conclusions

In summary, a complete set of Fmoc-protected nucleobase 1,2,3-triazole-linked amino acid derivatives (1,4-TzlNBAs) has been synthesized using click chemistry (CuAAC). These monomers were suitable for the solid-phase synthesis of a new type of nucleopeptides, HalTzls. Interaction studies using circular dichroism and fluorescence anisotropy have demonstrated that the designed and synthesized HalTzl ligands bind to complementary sequences of TAR RNA HIV-1, albeit with moderate affinity. This suggests that introducing additional positively charged amino acid residues, such as Lys and Arg, into the structure of HalTzls ligands could, through extra electrostatic interactions, enhance the affinity of HalTzls for TAR RNA. This enhancement may contribute to the development of new inhibitors blocking the TAR RNA-dependent transactivation during HIV replication.

## Data Availability

Data are contained within the article and Appendix A.

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
