# Peer review of "Solid Phase Synthesis and TAR RNA-Binding Activity of Nucleopeptides Containing Nucleobases Linked to the Side Chains via 1,4-Linked-1,2,3-triazole"

_biomedicines, 2024, doi:10.3390/biomedicines12030570_

Round 1

Reviewer 1 Report

Comments and Suggestions for Authors

This is a work on a novel nucleopeptide family that was synthesised by means of a click chemistry strategy and assayed in RNA binding experiments by CD and fluorescence. It must be revised to address the following major points:

abstract: no need of highlighting Nucleopeptides as bold. Moreover,  explain Hal already here . The sentence: This study demonstrates that newly designed HalTzls effectively bind TAR RNA structure, presenting a potential new class of RNA binders' lack the ultimate biomedical application, for future antiviral strategies? In a journal named Biomedicines I would expect reading the potential therapeutic utility of the presented molecules already in the abstract.

-line 96 Introduction: improved properties with respect to PNA are seeked in terms of solubility and absence of precipitation, a major limit of PNA that NPs can overcome. Discuss it and mention more works of prof. Domenica Musumeci from University of Naples on nucleopeptides in this respect

-line 115: Nucleopeptides were reported to show potential anti-HIV activity not only recognising RNAs essential to virus life, but also thanks to direct effects on HIV RT . The authors should discuss it in the light of the appropriate scientific literature

-line 151: also adenine has a free NH2 moiety. Why did you use it as unprotected molecule?

-line 169: you mention chirality in Fmoc-L-Aha but then 13 in Fig 4 and in general all NBAs and NPs are represented without specifying chirality in all figures. Why? Please check and correct

-Figure 6 can be part of Supporting Information

-Fig 8: provide a better quality Fig 8

-line 260:  Figure 7B, D should be  Figure 8B, D

-line 263: indicate the standard deviation values of the reported Tms

-line 274: why  a 10-fold excess? Similar to other literature reports? Cite them

-can you show in SI the CD of your NPs alone overlayed with RNA CDs?

-line 292: Figure 8 A, B should be Figure 9 A, B. Check also the subsequent references to the Figures

-Table 1: add the DeltaTm values for a quicker visualization of the effects of the ligands on RNA stabilities

-line 356: adding heterocyclic moieties to NPs structures does provide aspecific intermolecular interactions within multiple NP units that make DNA or RNA binding uneffective.  Consider in future to synthesise other NPs without triazole moieties.

-this was actually done by Musumeci D in RSC Adv 2016 and the RNA binding of a mixed sequence NPs was proven to be effective and specific. 

Author Response

Dear Reviewer,

thank you very much for the time and effort dedicated to the revision of our manuscript “Solid phase synthesis and TAR RNA-binding activity of nucleopeptides containing nucleobases linked to the side chains via 1,4-linked-1,2,3-triazole” by Piotr Mucha†&#*, Małgorzata Pieszko†$#, Irena Bylińska£, Wiesław Wiczk£, Jarosław Ruczyński†, Katarzyna Prochera†, Piotr Rekowski†& (Biomedicines-2809758). The manuscript has been deeply revised according to all the comments and valuable suggestions received. This document consists of a detailed, point-by-point response to each of these comments. The changes made in the manuscript have been highlighted in red colored text.

This is a work on a novel nucleopeptide family that was synthesised by means of a click chemistry strategy and assayed in RNA binding experiments by CD and fluorescence. It must be revised to address the following major points:

abstract: no need of highlighting Nucleopeptides as bold. Done. This was not intentional. It was an editing flaw in the writing of the manuscript.

Moreover,  explain Hal already here . Done. p1, l 16

 The sentence: This study demonstrates that newly designed HalTzls effectively bind TAR RNA structure, presenting a potential new class of RNA binders' lack the ultimate biomedical application, for future antiviral strategies? In a journal named Biomedicines I would expect reading the potential therapeutic utility of the presented molecules already in the abstract.  Done. p1, l 38.

-line 96 Introduction: improved properties with respect to PNA are seeked in terms of solubility and absence of precipitation, a major limit of PNA that NPs can overcome. Discuss it and mention more works of prof. Domenica Musumeci from University of Naples on nucleopeptides in this respect.  Done p3, l 100-116   I have added some new information about the advantage of NPs over PNAs. I have also added suggested literature.

-line 115: Nucleopeptides were reported to show potential anti-HIV activity not only recognising RNAs essential to virus life, but also thanks to direct effects on HIV RT . The authors should discuss it in the light of the appropriate scientific literature

I have also included the suggested inhibitory effect of NPs on HIV RT activity along with suggested literature. However, I do not want to have a discussion on NP-dependent RT inhibition, as the main topic of the manuscript is RNA/TAR recognition by NPs.

-line 151: also adenine has a free NH2 moiety. Why did you use it as unprotected molecule?

We were aware that the NH2 group of adenine remained unprotected. The coupling of the free electron pair of the nitrogen atom to the electron decet of the purine system results in little reactivity to the a amine group of the NBA monomers. We monitored the behavior of this unprotected aromatic amine group during alkylation and during acylation during the synthesis of HalTzls. In both cases, this group was chemically passive under the synthesis conditions used. ESI MS analysis of the crude products of these reactions did not reveal the presence of by-products formed with its participation. Therefore, we left the NH2 group in the adenine derivative of NBAs unprotected during the synthesis of NPs.

-line 169: you mention chirality in Fmoc-L-Aha but then 13 in Fig 4 and in general all NBAs and NPs are represented without specifying chirality in all figures. Why? Please check and correct  Done

Yes, Fmoc-L-Aha was a precursor to the syntheses of NBAs and HalTzls presented in our work. Since none of the reactions performed changed the configuration of the asymmetric carbon atom, we did not consider the issue of stereochemistry on the presented structures. However, changes to take into account the stereochemistry of the obtained structures were made to Figure 1 and Scheme 3. I have also included additional information about configuration retention on the alpha carbon in the manuscript (p9 l 364). Since the intention of Scheme 4 is to present the synthesis of HalTzls, the NPs structures were left unchanged.

-Figure 6 can be part of Supporting Information

I have moved figure 6 to SI. Although, on the other hand, I think that from the point of view of someone who is involved in synthesis using compounds containing a cytosine protected by an acetyl group, the information ( and the chromatograms) presenting the fact that under the standard conditions for cleaving the peptide from the resin, the protecting group is only partially removed and an additional procedure is necessary to allow its complete removal may be a valuable information.

-Fig 8: provide a better quality Fig 8  Done

-line 260:  Figure 7B, D should be  Figure 8B, D  Done

-line 263: indicate the standard deviation values of the reported Tms   Done

-line 274: why  a 10-fold excess? Similar to other literature reports? Cite them

OK, you are right the use of 10 times excess ligand-NPs is not the standard stoichiometry of CD studies. Typically, slightly smaller excesses are used. However, knowing the previous moderate Kd values from fluorescence studies and having a limited amount of labeled TAR and NPs, we decided to use a 10-fold excess of NPs.

-can you show in SI the CD of your NPs alone overlayed with RNA CDs?

Unfortunately, we can't do that now. Since the NPs studied are short (3- and 6-mer) we did not expect the CD spectra to show anything interesting related to their conformation. Therefore, we did not record CD spectra of the free NPs. When performing CD experiments, the NPs contribution was automatically subtracted from the ellipticity effect obtained for the TAR/NP complex. At the moment we do not have enough free NPs to perform CD spectra for them.

-line 292: Figure 8 A, B should be Figure 9 A, B. Check also the subsequent references to the Figures Done

-Table 1: add the DeltaTm values for a quicker visualization of the effects of the ligands on RNA stabilities Done

-line 356: adding heterocyclic moieties to NPs structures does provide aspecific intermolecular interactions within multiple NP units that make DNA or RNA binding uneffective. 

In my opinion, it is not that simple and obvious. The general view is that the introduction of a triazole residue into the structure of a compound "improves its properties". Whatever that means. On the one hand, the triazole ring may generate the appearance of additional non-specific intermolecular interactions, e.g. of the hydrogen bonding type with hydroxyl groups of ribose residues and/or heavy atoms of nitrogenous bases (e.g. NH, NH2) in RNA or hydrophobic interactions, e.g. of the base stacking type with nitrogenous bases. On the other hand, the introduction of additional Lys and Arg residues into the NP sequence also generates the appearance of non-specific electrostatic interactions with nucleic acids, which increase their affinity. If one does not exaggerate their number, the resultant effect is positive. What this looks like in the case of the introduction of a triazole system into the structure of a NP, we do not know. Our research is only just beginning to explore this issue. So far, NPs with triazole systems have not been synthesized and their interactions with unique RNA structures, which are important for viral replication, have not been studied.  I have raised the aspect of the impact of the triazole ring on DNA/RNA structure in an extended version of the Results and Discussion section

Consider in future to synthesize other NPs without triazole moieties.

-this was actually done by Musumeci D in RSC Adv 2016 and the RNA binding of a mixed sequence NPs was proven to be effective and specific. 

Yes, we are thinking of synthesizing NPs corresponding to those presented in the manuscript but lacking the triazole system. However, in terms of synthesis requirements, this is a much more difficult task than Prof. Musumeci's approach (RSC Adv 2016). Attaching an acetyl derivative of a nucleobase to an amino group in the side chain  through classical methods of peptide bond synthesis is much easier than introducing a nucleobase using an alkylation reaction.

Reviewer 2 Report

Comments and Suggestions for Authors

The paper deals with the solid-phase synthesis of three nucleopeptides with the aim to complement bulge and outer loop structures of TAR RNA HIV-1. The solid-phase synthesis method is straightforward, while the CD study revealed moderate binding affinity. The authors finally propose the insertion of Lys/Arg amino acids to have better results.

One basic limitation of this work it that only three structures were tested, while some modifications are suggested in the end. There is no designing method/approach or thorough screening of compounds to allow a well-established proposal for the future directions of this study. At least, the suggested structures (with Lys/Arg) should have been part of this work. At this point, this is not an important, but rather incomplete contribution.

Moderate editing of English language is required, especially in the synthetic part. Further difficulties of the authors in the chemical part of the paper in the results and discussion section is noticed, where significant presentation/discussion problems exist. At this point this is just a quick explanation of the concept and a quick description of the figures/schemes. On the other hand, the CD study is better presented/described. Some other comments are the following:

1.    The stereochemistry of the amino acids should be drawn.

2.     The triple bonds should be linear (sp hybridized)

3.    The description in the Figures/Schemes should be limited to the image/scheme content. Any other description that is not part of the Figure /Scheme should be incorporated in the main text. For example, in Figure 1, lines 137-139 should not be part of the Figure legend.

4.    Some of the Figures that describe synthesis should be named Schemes.

5.    In Figure 2, the exact chemical reaction requires BH in the reactants, in order to have B in the products.

6.    In Figure 5, the resin functional groups involved in amino acid attachment should be shown; reactants, conditions, etc should also be indicated.

7.    In Figure 6, the hplc of all three final products (analytical hplc) should be shown.

8.    The yields of the reactions in solution should be indicated in the schemes (under the chemical reaction arrow).

9.    On line 201, table 1 refers to table 3. Renumbering of the tables is needed.

Comments on the Quality of English Language

Moderate editing of English language is required, especially in the synthetic part. Further difficulties of the authors in the chemical part of the paper in the results and discussion section is noticed, where significant presentation/discussion problems exist. At this point this is just a quick explanation of the concept and a quick description of the figures/schemes. 

Author Response

Dear Reviewer,

thank you very much for the time and effort dedicated to the revision of our manuscript “Solid phase synthesis and TAR RNA-binding activity of nucleopeptides containing nucleobases linked to the side chains via 1,4-linked-1,2,3-triazole” by Piotr Mucha†&#*, Małgorzata Pieszko†$#, Irena Bylińska£, Wiesław Wiczk£, Jarosław Ruczyński†, Katarzyna Prochera†, Piotr Rekowski†& (Biomedicines-2809758). The manuscript has been deeply revised according to all the comments and valuable suggestions received. This document consists of a detailed, point-by-point response to each of these comments. The changes made in the manuscript have been highlighted in red colored text.

The paper deals with the solid-phase synthesis of three nucleopeptides with the aim to complement bulge and outer loop structures of TAR RNA HIV-1. The solid-phase synthesis method is straightforward, while the CD study revealed moderate binding affinity. The authors finally propose the insertion of Lys/Arg amino acids to have better results.

One basic limitation of this work it that only three structures were tested, while some modifications are suggested in the end. There is no designing method/approach or thorough screening of compounds to allow a well-established proposal for the future directions of this study. At least, the suggested structures (with Lys/Arg) should have been part of this work. At this point, this is not an important, but rather incomplete contribution.

In the next stage of the research, we plan to synthesize mixed NPs containing basic amino acid residues (Lys, Arg).  Our intention was to start with NP structures containing only 1,4TzlNBAs, without additional structural elements such as Lys or Arg. The project proved to be challenging due to the lack of commercial availability of reactants (with the exception of Fmoc-Aha). It took us more than two years to synthesize a set of NBAs and three HalTzls. Therefore, we decided to publish the results as presented in the manuscript.

Moderate editing of English language is required, especially in the synthetic part. Further difficulties of the authors in the chemical part of the paper in the results and discussion section is noticed, where significant presentation/discussion problems exist. At this point this is just a quick explanation of the concept and a quick description of the figures/schemes. On the other hand, the CD study is better presented/described.

Section Results and Discussion has been corrected and extended.

Some other comments are the following:

  1. The stereochemistry of the amino acids should be drawn. Done
  2. The triple bonds should be linear (sp hybridized) Done
  3. The description in the Figures/Schemes should be limited to the image/scheme content. Any other description that is not part of the Figure /Scheme should be incorporated in the main text. For example, in Figure 1, lines 137-139 should not be part of the Figure legend. Done
  4. Some of the Figures that describe synthesis should be named Schemes. Done
  5. In Figure 2, the exact chemical reaction requires BH in the reactants, in order to have B in the products. Done
  6. In Figure 5, the resin functional groups involved in amino acid attachment should be shown; reactants, conditions, etc should also be indicated. Done
  7. In Figure 6, the hplc of all three final products (analytical hplc) should be shown. Done. Figure has been moved to Supplement material section at the request of one of the reviewers
  8. The yields of the reactions in solution should be indicated in the schemes (under the chemical reaction arrow). Done
  9. On line 201, table 1 refers to table 3. Renumbering of the tables is needed. Done

Comments on the Quality of English Language

Moderate editing of English language is required, especially in the synthetic part. Further difficulties of the authors in the chemical part of the paper in the results and discussion section is noticed, where significant presentation/discussion problems exist. At this point this is just a quick explanation of the concept and a quick description of the figures/schemes. 

Some correction of synthetic part of the manuscript have been made. Section Results and Discussion has been corrected and extended.

Reviewer 3 Report

Comments and Suggestions for Authors

This study showed a potential new class of RNA binders, i.e., newly designed HalTzls binding to TAR RNA structure. The article language is readable, even if a minor editing of English is required. TAR, both in the abstract and introduction section, should be written in the extended form and not as acronym to allow the reader to understand. Figures quality should be improved. Statistics is missing and should be added to graphs. Finally, I have a last concern: why did authors analyse TAR RNA in HIV-1? It should be described in the text.

Comments on the Quality of English Language

Minor editing is required, e.g. typing error such as an unique and not a unique (lane 24)

Author Response

Dear Reviewer,

thank you very much for the time and effort dedicated to the revision of our manuscript “Solid phase synthesis and TAR RNA-binding activity of nucleopeptides containing nucleobases linked to the side chains via 1,4-linked-1,2,3-triazole” by Piotr Mucha†&#*, Małgorzata Pieszko†$#, Irena Bylińska£, Wiesław Wiczk£, Jarosław Ruczyński†, Katarzyna Prochera†, Piotr Rekowski†& (Biomedicines-2809758). The manuscript has been deeply revised according to all the comments and valuable suggestions received. This document consists of a detailed, point-by-point response to each of these comments. The changes made in the manuscript have been highlighted in red colored text.

This study showed a potential new class of RNA binders, i.e., newly designed HalTzls binding to TAR RNA structure. The article language is readable, even if a minor editing of English is required.

Some linguistic corrections have been made to the manuscript.

TAR, both in the abstract and introduction section, should be written in the extended form and not as acronym to allow the reader to understand.  Done

Figures quality should be improved. Done.

Statistics is missing and should be added to graphs. At present, I am only able to include statistical aspects taking into account the standard deviation value provided in the relevant graphs and tables.

Finally, I have a last concern: why did authors analyze TAR RNA in HIV-1? It should be described in the text. It is. P3, l 132-140, l 144-146.

Round 2

Reviewer 1 Report

Comments and Suggestions for Authors

The paper can be accepted in the current form 

Reviewer 2 Report

Comments and Suggestions for Authors

Although the work is limited to a few peptide structures, the positive management as well as the explanations given are positively evaluated. In addition, all comments/suggestions were addressed.